# Subungual Exostosis Presenting as a Pyogenic Granuloma-like Lesion with Reactive Myofibroblastic Proliferation in Two Young Women

Rosanna Fox [1], Effie Katsarma [2], Nick Tiffin [3] and Manuraj Singh [3,*]

1   Department of Acute Medicine, West Middlesex University Hospital, London TW7 6AF, UK;
    rfox@doctors.org.uk
2   Department of Plastic and Reconstructive Surgery, Chelsea and Westminster Hospital,
    London SW10 9NH, UK; ekatsarma@msn.com
3   Department of Dermatopathology, St George's University Hospital, London SW17 0QT, UK;
    ntiffin72@gmail.com
*   Correspondence: masingh@doctors.org.uk

**Abstract:** Subungual exostosis (SE) is a well-recognised benign proliferation of the distal phalanx most often seen in young adults and affecting the big toe. Possible triggers include previous trauma and chronic irritation or infection. We describe two atypical cases of SE in two young women presenting with pyogenic granuloma-like lesions clinically. Diagnostic biopsies were performed to confirm the diagnosis and excluded amelanotic melanoma. However, histology unexpectedly revealed reactive myofibroblastic proliferations mimicking nodular fasciitis overlying the SE. Given the atypical clinical presentation, the diagnosis was initially missed or not considered in both patients. They highlight two important points; the first is that SEs may present with pyogenic granuloma-like lesions clinically and that histological analysis is then required to exclude malignancy, particularly amelanotic melanoma. Secondly, that the histology will show a reactive myofibroblastic proliferation and if the sample is relatively superficial and pathologists are not aware of this potential reaction pattern, the underlying diagnosis of SE may be missed.

**Keywords:** subungual exostosis; reactive myofibroblastic proliferation; pyogenic granuloma; nodular fasciitis; amelanotic melanoma

## 1. Introduction

Subungual exostosis (SE) is a solitary benign osteocartilaginous tumour of the distal phalanx first described by Dupuytren in 1847 [1,2]. It may affect any finger or toe but up to 80% of cases occur in the big toe [3]. In typical cases, as the SE increases in size, it will collide with the overlying nail plate, causing swelling and discomfort and pushing the nail upwards which eventually detaches from the underlying bed. Clinically, a firm fixed nodule with a hyperkeratotic surface is seen [1]. Plain film radiographs will show a radio-opaque mass on the dorsomedial side of the distal phalanx, although this will lack contiguity with the medullary cortex, distinguishing it technically from an osteochondroma.

Given its potential to simulate other subungual tumours, SE may be misdiagnosed, and histology and plain X-ray should help differentiate SE from other lesions, especially malignancies [4]. SE has a fibrocartilaginous cap histologically, whereas osteochondroma has a hyaline cartilage confluent with the bone cortex [4,5]. The overlying lesion may present clinically as a skin coloured or erythematous nodule and, therefore, mimic lesions such as amelanotic melanoma, glomus tumour, verruca vulgaris, or as in the cases described here, pyogenic granuloma [6]. In many cases, where there is misdiagnosis or a significant delay in diagnosis, patients may experience significant pain and onychodystrophy, affecting their quality of life. The mainstay of treatment is marginal surgical excision of the exostosis

and wound closure, which has good cosmetic and functional results and a low recurrence risk [2,7].

We describe two cases both in young females where the lesion presented as an ulcerated erythematous nodule mimicking a pyogenic granuloma clinically and, therefore, also raised the possibility of a subungual amelanotic melanoma and necessitating a diagnostic tissue biopsy. The diagnosis of SE was subsequently confirmed by plain X-rays. Interestingly, the histology from the overlying erythematous nodule in both cases showed a reactive myofibroblastic proliferation with features overlapping with nodular fasciitis. We highlight this reactive myofibroblastic proliferation to pathologists as a potential sign of underlying SE. This is of particular relevance as biopsies from the nail bed area tend to be relatively superficial and therefore may not include the diagnostic features of underlying bone and cartilage. In addition, and as demonstrated by our cases, clinicians may not originally consider SE in the differential diagnosis of a pyogenic granuloma-like lesion affecting the nail bed, and as a result of this, a diagnostic plain X-ray will not be ordered. Therefore, the diagnosis may be missed if clinicians and/or pathologists are not aware of this potential clinicopathological presentation.

## 2. Case Reports

Case 1: A 24-year-old, otherwise healthy female reported multiple recurrent episodes of an apparent ingrown toenail affecting her right big toe. She underwent surgical intervention on the nail by a podiatrist in December 2018 and after recurrence proceeded to have a partial nail avulsion and nail bed ablation three months later. She had a further recurrence and developed a large lesion covering the majority of the nail bed area and was referred to dermatology in June 2019. There was a history of wearing tight fitting high heeled shoes. On examination, there was a large erythematous nodule covering almost the entire nail bed area of the right big toe (Figure 1).

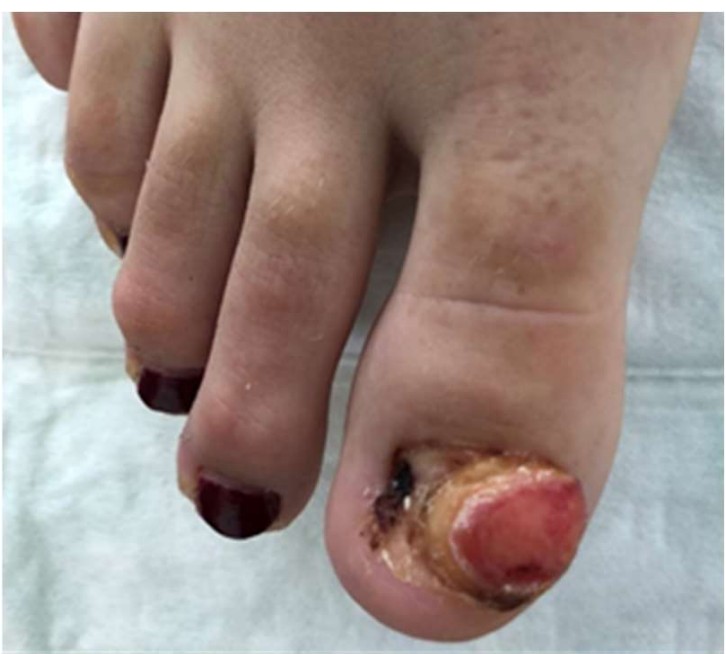

**Figure 1.** Ulcerated erythematous nodule covering the majority of the nail bed of the right big toe.

It measured approximately 20 × 20 mm. At this stage a SE was suspected, and a plain X-ray showed a significant exostosis arising from the distal phalanx of the toe (Figure 2). The lesion was shaved and sent for histology and a second specimen was taken from the bone at the deep margin of the lesion. The histology showed a reactive myofibroblastic proliferation with a myxoid stroma and focal areas of keloidal-like collagen overlying mature cartilage and bone fully in keeping with a SE (Figure 3A–D). The deep specimen showed fragments of

mature bone. Immunohistochemistry confirmed the myofibroblastic phenotype overlying the SE with the majority of spindle cells showing strong and diffuse smooth muscle actin (SMA) positivity (Figure 3E). There was also so called "tram-track" morphology to the SMA staining pattern which is very suggestive of myofibroblasts (Figure 3F). The spindle cells were negative for AE1/3, MNF116, S100, CD34, p63, and desmin. The exostosis was fully removed in July 2019 and the defect was covered with a full-thickness skin graft from the right groin. The post-operative period was uneventful and follow-up showed good donor and graft site healing and no signs of recurrence. Histology of the final surgery showed an ulcerative SE with fibrous granulation tissue representing reactive changes to the previous surgical intervention.

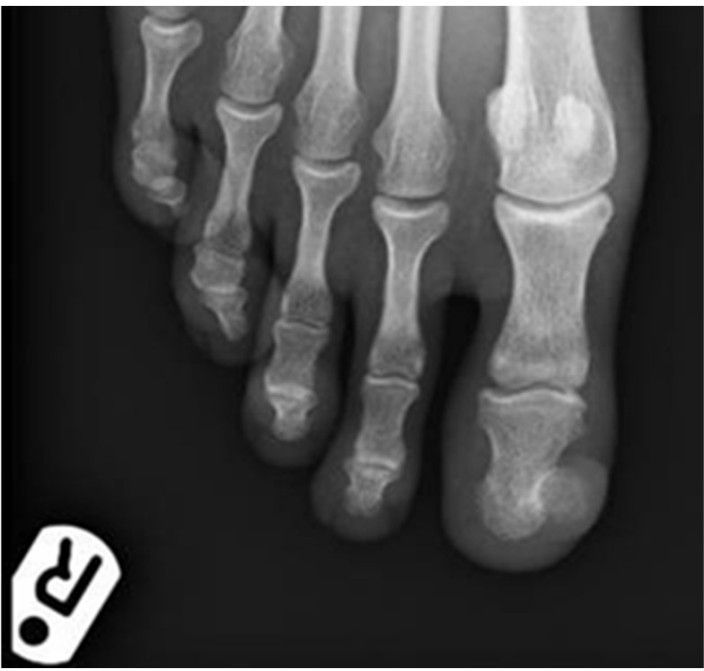

**Figure 2.** Antero-posterior plain radiograph of the right forefoot showing exostosis arising from the distal phalanx of the right big toe.

Case 2: A 20-year-old female long-distance runner presented with a five-week history of a lesion over the dorsal aspect of her right big toe. She was a university athletic scholar and reported wearing tight fitting spike running shoes to train. She was originally seen by another dermatologist who was concerned about a subungual amelanotic melanoma and referred the patient for biopsy to our department. On examination, there was a subungual erythematous nodule covering the majority of the right big toenail bed (Figure 4). Because of our previous experience with Case 1, a SE was suspected at the same time a diagnostic biopsy was being performed to exclude amelanotic melanoma. As a result, a plain X-ray showed an exostosis of the distal phalanx and histology showed ulcerated tissue with an underlying myofibroblastic proliferation associated with myxoid stroma and focal keloidal-like collagen (Figure 5A–C). Of note, this sample was relatively superficial and, therefore, did not show underlying cartilage and bone. Immunohistochemistry showed that the lesional spindle cells were strongly and diffusely positive for SMA (Figure 5D) and negative for keratin and melanocytic markers. As with the first case, this suggested a reactive myofibroblastic proliferation secondary to the underlying exostosis. The patient decided not to proceed with surgery given her concerns regarding this impacting her athletic training and performance.

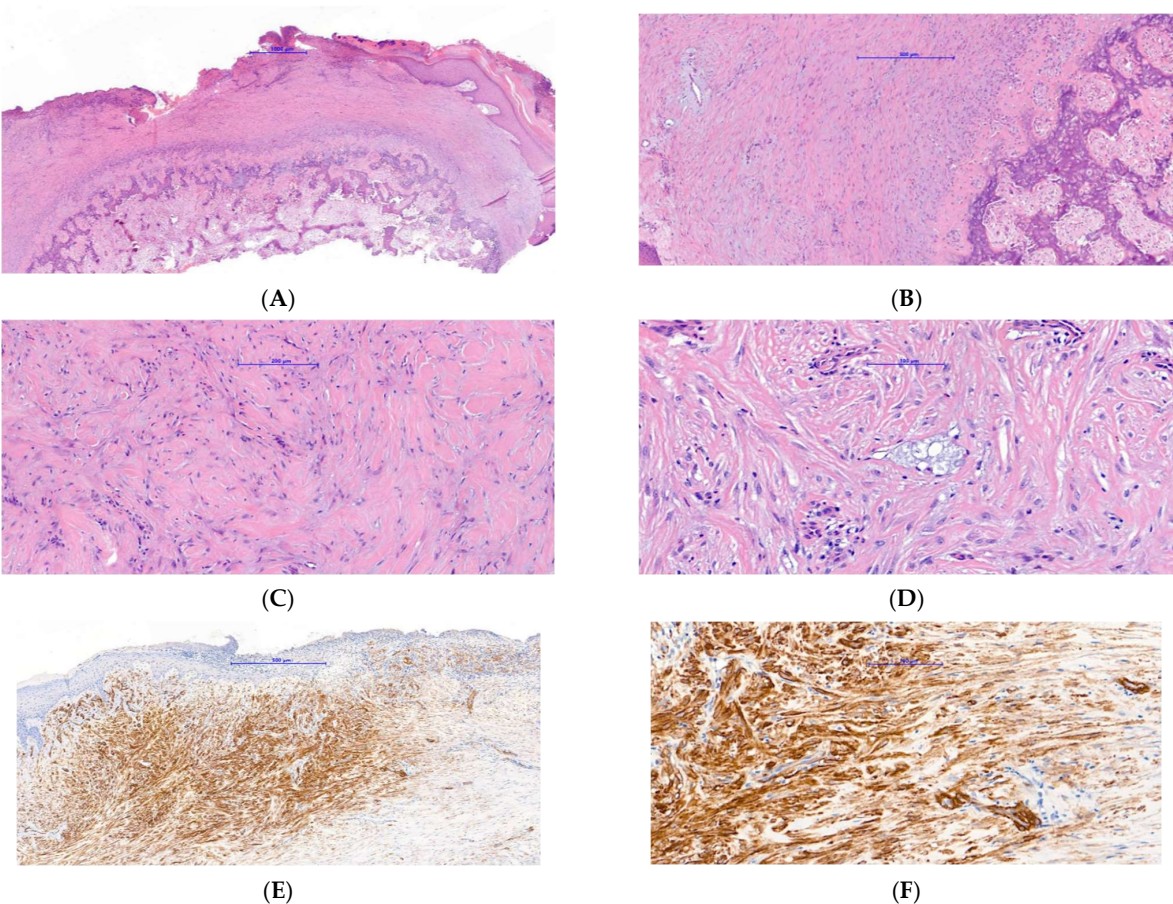

**Figure 3.** Histopathology of case 1: (**A**) low power image showing an ulcerated spindle cell proliferation overlying cartilage and bone. (**B**) Medium power image showing intimate relationship of spindle cell proliferation with underlying cartilage and bone. (**C**) High power image of spindle cell proliferation demonstrating a myofibroblastic morphology associated with myxoid stroma and focal keloidal-like collagen. (**D**) High power image highlighting focal microcysts of myxoid stroma reminiscent of those seen in nodular fasciitis. (**E**,**F**) low and high power images, respectively, of immunohistochemistry for SMA, showing strong positivity of the spindle cell proliferation with a "tram track" pattern of staining fully in keeping with a myofibroblastic phenotype.

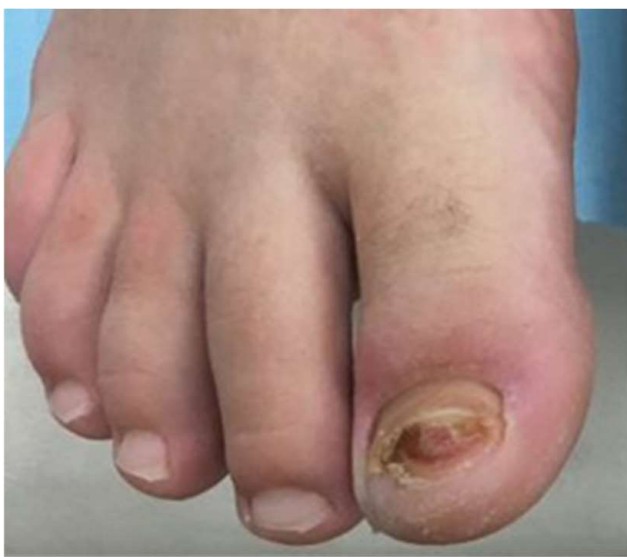

**Figure 4.** Subungual erythematous nodule covering the majority of the right big toenail bed.

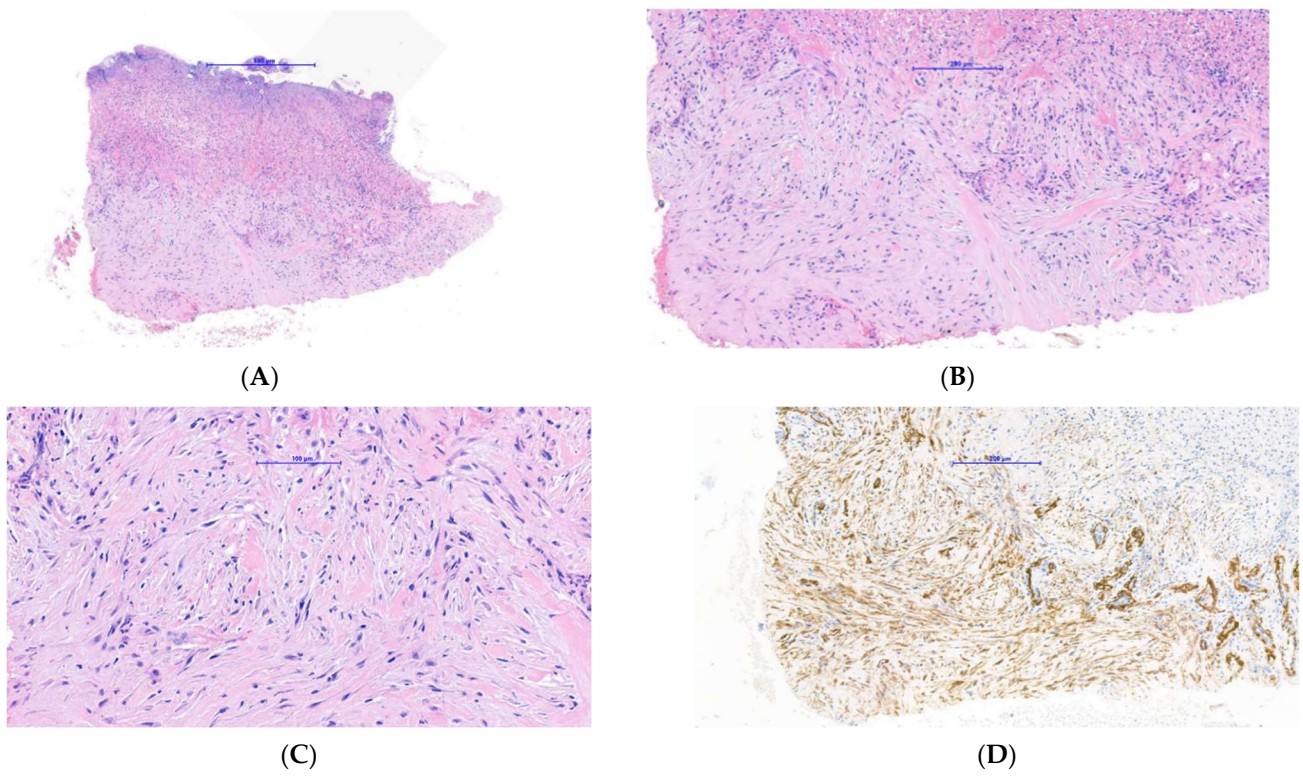

**Figure 5.** Histopathology of case 2: (**A**) Low power image showing an ulcerated spindle cell proliferation. (**B**) Medium power image showing spindle cell proliferation associated with myxoid stroma and focal keloidal-like collagen. (**C**) High power image of spindle cell proliferation demonstrating a myofibroblastic morphology and highlighting focal areas of red cell extravasation and hemosiderin deposition. (**D**) Immunohistochemistry for SMA, showing strong and diffuse positivity of the spindle cell proliferation in keeping with a myofibroblastic phenotype.

## 3. Discussion

SEs are benign bony tumours located beneath the nail bed on the dorsal aspect of the distal phalanx. The majority of lesions (up to 80%) occur on the distal phalanx of the hallux (big toe), as in the cases described here [8]. There are reports in the literature of SEs occurring on the hand, but these are relatively rare [9]. SEs typically arise during the second or third decades of life and affect females more commonly than males [10]. The exostosis begins as an area of fibrocartilaginous metaplasia beneath or adjacent to the nail bed, which then undergoes endochondral ossification and conversion to trabecular bone, surrounded by a fibrocartilaginous cap [8,10]. As the tumour matures, this fibrocartilaginous cap will eventually fuse the nail bed to the distal phalanx beneath [8]. This results in digital discomfort in weight-bearing, onycholysis, and onychodystrophy. In ulcerating lesions, patients are also at an increased risk of secondary infection.

Clinically, SEs are inconsistent in their presentation and can mimic onychocryptosis (ingrown nail), keratoacanthoma, fibroma, hemangioma, epidermal cyst, paronychia, onychomycosis, verruca vulgaris, myositis ossificans, malignant tumours, and pyogenic granuloma [11,12]. SEs may present with ulceration, infection, and surrounding tissue changes meaning the underlying bony abnormality is often missed or not considered. The two cases we describe were initially thought to be pyogenic granuloma or amelanotic melanoma due to their clinical features, and underlying bone pathology was not initially considered. Given the breadth of differentials, the diagnosis of SE may be a challenge for dermatologists and other clinicians as we have demonstrated. Indeed, the prevalence of SEs may be underestimated due to frequent misdiagnoses [5]. As a result, it is important that SE is considered in any digital-tip pathology involving the nail.

There is no clearly established cause of SE although chronic infection, trauma, and chronic irritation are all suggested in the literature as causes of fibrocartilaginous metaplasia [1,13]. Both patients described in this paper had an association with tight fitting shoes; the patient in case 1 was a regular tight fitting, high heeled shoe wearer, and the athlete in case 2 regularly wore tight fitting studded shoes for training. It is likely that these tight fitting shoes caused chronic irritation or low-grade trauma to the distal phalanx, and this highlights the importance of shoe-wear history in patients presenting with nail lesions. LeninBabu et al. reported a case in which a patient who had undergone previous Zadek's procedure for onychocryptosis subsequently developed a SE [14]. The patient described in our first case also reported a similar nail bed intervention for presumed onychocryptosis which caused worsening of the lesion. Rarer cases of hereditary multiple exostoses and other related syndromes have also been shown to cause SEs [15,16].

Diagnosis of SE can be confirmed with plain radiography, which should ideally be performed before an invasive biopsy. In the patient described in our first case, an early X-ray could have prevented two surgical interventions on the nail and inevitable recurrence prior to definitive excision. Our two patients presented with erythematous nodules mimicking pyogenic granuloma clinically and, therefore, diagnostic biopsies were considered necessary in order to exclude subungual amelanotic melanoma. Interestingly, the histology of both patients showed unexpected reactive myofibroblastic proliferation with myxoid change and keloidal-like collagen. These changes with the addition of focal microcysts of myxoid stroma and focal red cell extravasation were somewhat reminiscent of nodular fasciitis in areas. One may consider that this myofibroblastic proliferation was secondary to previous surgical intervention in case 1; however, very similar histological findings were seen in case 2 which had no history of previous surgical intervention.

The differential diagnosis histologically includes other subungual myofibroblastic and fibroblastic proliferations such as cellular fibrous histiocytoma, fibromatosis, digital fibromyxoma, and nodular fasciitis. The first three can usually be differentiated by morphology and/or immunohistochemistry. For example, fibromatosis tends to have longer sweeping fascicles, cellular fibrous histiocytoma peripheral collagen entrapment, and digital fibromyxoma is usually CD34 positive. However, small biopsies may cause extreme difficulties. We feel that our cases overlapped most with nodular fasciitis, but there tends to be more red cell extravasation and myxoid microcyst formation in typical cases of nodular fasciitis. Another condition to consider in an acral location is fibro-osseous pseudotumour of digits. This is characterised by a fibroblastic/myofibroblastic proliferation in a myxoid stroma, intimately associated with immature bone trabeculae rimmed by osteoblasts. However, this lacks the typical zoning phenomenon seen in subungual exostosis usually allowing for easy distinction.

We feel that it is important to highlight this reactive myofibroblastic proliferation to pathologists as a clue to the potential presence of an underlying SE. This myofibroblastic proliferation being a potential sign of an underlying SE is not well described in the literature. In particular and as our cases demonstrated, SE may present as an erythematous nodule mimicking pyogenic granuloma clinically and therefore necessitating histological analysis to exclude a subungual amelanotic melanoma. Given the fact that nail bed biopsies may be relatively superficial, as in our case 2, it is likely in such circumstances that histological analysis alone will only reveal this reactive myofibroblastic proliferation without underlying cartilage and bone. With pathologists being made more aware of this reactive myofibroblastic proliferation being a potential sign of an underlying SE, they are more likely to highlight this possibility to the supervising clinician, who, as in our cases, may not have initially considered the correct diagnosis. We aim to highlight this potential trap to clinicians and pathologists to help prevent possible misdiagnoses and incorrect treatments.

The gold standard of treatment remains as a localised excision of the exostosis, which can often be performed with nail-sparing techniques to improve cosmetic outcomes and protect the underlying nail bed [1]. Incomplete excision of the lesion results in a recurrence rate of up to 50% [17]. There remains ongoing debate regarding the most effective timing

of surgical excision [8,17]. Some have suggested early intervention to prevent secondary infection, relieve discomfort, and prevent total onychodystrophy [17]. While others report that delaying excision until the lesion reaches maturity, allows a cleavage plane to develop between the fibrocartilaginous cap and the overlying nail bed, minimising the risk of recurrence [18].

In summary, our two cases of SEs highlight a characteristic, but until now, not well described clinicopathological presentation. They show that a SE may present as an erythematous nodule mimicking a pyogenic granuloma clinically and that histology depending on the depth of the biopsy will typically show a reactive myofibroblastic proliferation with or without underlying cartilage and bone. They also highlight the importance of early plain radiography and the potential diagnostic trap of a superficial biopsy for the reporting pathologist, especially when the lesion is not clinically suspected. Given the morbidity and quality of life outcomes associated with a late or incorrect diagnosis, being aware of this characteristic clinicopathological scenario is important to help avoid misdiagnosis of SEs.

**Author Contributions:** Conceptualization, M.S.; methodology, M.S. and R.F.; software, M.S. and R.F.; validation, M.S., R.F., E.K. and N.T.; formal analysis, M.S. and R.F.; investigation, M.S., E.K. and N.T.; resources, R.F.; data curation, M.S. and R.F.; writing—original draft, R.F.; writing—review and editing, M.S. and R.F.; visualization, M.S. and R.F.; supervision, M.S.; project administration, M.S. and R.F. All authors have read and agreed to the published version of the manuscript.

**Funding:** This research received no external funding.

**Institutional Review Board Statement:** Not applicable.

**Informed Consent Statement:** Written informed consent has been obtained from the patients to publish this paper.

**Conflicts of Interest:** The authors declare no conflict of interest.

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
