# Peer review of "Subungual Exostosis Presenting as a Pyogenic Granuloma-like Lesion with Reactive Myofibroblastic Proliferation in Two Young Women"

_dermatopathology, doi:10.3390/dermatopathology9020024_

Round 1
Reviewer 1 Report
1. The resemblance to pyogenic granuloma in these two cases is clinical and not pathologic. I would clarify this in appropriate areas of the manuscript.
2. The authors should expand on the differential diagnosis of a subungual myofibroblastic proliferation. In other words, without the benefit of an x-ray, what other considerations should be listed in a pathology report for these two specimens?
Author Response
Many thanks for your comments. Please see the revised attached document.
- We have now used the word 'clinically' in lines 18, 38, 45, 165, 190 and 210 as highlighted in the text which should ensure the resemblance to pyogenic granuloma is clarified as clinical rather than pathologic.
- We have highlighted a new paragraph in lines 173-185 which expands on the differential diagnoses of subungual myofibroblastic and fibroblastic proliferations.

Reviewer 2 Report
This is an interesting observation. The clinical photos are excellent. My main concern is about case 2 as in your photomicrographs you don't have the included features of an underlying subungual exostosis. I see that you did mention this in your text, and I see that you do mention the x ray findings for this patient. I think it is important to be able to see the histopathological relationship between the subungual exostosis and the myofibroblastic proliferation. I understand the patient did not want additional surgery. However, from an academic point of view, I would recommend removing case 2 from the manuscript as it is just not a good example of what you are describing. Perhaps you can search your files for a better example and substitute it. People may not have been aware to think about this interesting finding you describe. The manuscript still works well with the excellent case 1 description.
Author Response
Thank you for your helpful comments. Unfortunately we do not have another example of this phenomenon in our patient database. We do feel that case 2 nicely highlights the high potential for missing this diagnosis with superficial biopsies that are commonly performed at acral sites. We do feel the paper benefits from a second case so as to further support this probably under recognised phenomenon.